# Exploring healthcare professionals' beliefs, experiences and opinions of family-centred conversations when a parent has a serious illness: A qualitative study

Louise J. Dalton[1]*, Abigail McNiven[2], Jeffrey R. Hanna[1,3], Elizabeth Rapa[1]

1 Medical Sciences Division, Department of Psychiatry, University of Oxford, Oxford, United Kingdom,
2 Medical Sciences Division, Nuffield Department of Primary Care Health Sciences, University of Oxford, Oxford, United Kingdom, 3 School of Nursing and Midwifery, Queen's University Belfast, Belfast, United Kingdom

* louise.dalton@psych.ox.ac.uk

## Abstract

This study explored healthcare professionals' perceived role in talking to adult patients about sharing their diagnosis with children. Semi-structured interviews were conducted to explore healthcare professionals' beliefs about how families could and should be supported when a parent has a serious illness. Participants were 24 healthcare professionals working in primary, secondary and tertiary NHS services in the UK with adult patients diagnosed with a serious illness. Data were analysed thematically. Many healthcare professionals reported systems to identify patients' family relationships, but this information was rarely used to initiate conversations on what and how to talk to children. It was frequently assumed that someone else in the healthcare system was supporting patients with family communication. Others reported there were more urgent priorities for the consultation or considered that talking to children was a private family matter. However, several professionals did undertake these conversations, viewing this as a central part of their role. Some healthcare professionals felt they had inadequate skills or confidence to raise talking to children with their patients and indicated a need for specific training to address this. The results highlight the importance of systematically documenting patients' relationships with children so that this information can be used to inform ongoing discussions with the healthcare team about what children have been told. Patients consistently report wanting support about how to talk to children and the benefits of effective communication are well documented. Dissemination of this evidence could encourage professionals across all specialities to include family-centred communication in routine patient care. Training resources are needed so that staff feel empowered and equipped to raise these sensitive subjects with their patients.

**Data Availability Statement:** Data cannot be shared publicly as the data contains potentially identifiable and sensitive patient information. The Central University of Oxford Ethics Committee

(CUREC) (Reference R61317) only granted ethical approval for the data to be shared within the named research team (authors) and authorised representatives from the University of Oxford for monitoring and/or audit of the study to ensure compliance with regulations. CUREC can be contacted via email curec@admin.ox.ac.uk.

**Funding:** ER, LJD and AM received funding from the University of Oxford John Fell Fund Grant number 163/111 https://researchsupport.admin.ox.ac.uk/funding/internal/jff JRH received funding from the Westminster Foundation https://westminsterfoundation.org.uk/ The funders had no role in considering the study design or in the collection, analysis, interpretation of data, writing of the report or decision to submit the article for publication.

**Competing interests:** The authors have declared that no competing interests exist.

# Introduction

Effective communication with children about parental illness has long term benefits for children and their family's physical and mental health [1, 2]. These include better child psychological well-being, as well as lower reported symptoms of depression, anxiety and behavioural problems [1, 3]. Furthermore, communication with children about a parent's diagnosis has benefits for parental mental health, treatment adherence and family functioning [4–6]. Although the benefits of talking to children about a loved one's illness are well documented [7], parents find sharing a diagnosis with their children one of the most difficult experiences [8]. Understandably, parents want to protect their children from distress and many feel uncertain how to navigate these sensitive conversations. Parents want help and advice from their healthcare team about how to talk to children about a parent's diagnosis, prognosis and treatment, yet often report finding this support difficult to obtain [9, 10]. Consequently, parents may avoid talking about the illness of themselves or a significant adult with their children [11, 12]. However, children are "astute observers" and are often acutely aware of changes within the family [1]. Parental silence about what is happening risks children misinterpreting the situation and worrying alone, without access to the emotional support they need [4, 13].

Healthcare professionals (HCPs) consistently describe the challenges of talking with patients about communicating news of their illness with children [14]. HCPs report concerns that starting conversations with patients about their children will not be welcomed or create distress for the patient. These conversations may cause sadness and fear for the HCP themselves [15]. The ever-increasing pressure of time on appointments is a further challenge for family-centred conversations with patients [16]. HCPs report a lack of adequate knowledge about children's developmental understanding or age-appropriate communication to have conversations with patients about their children [17]. HCPs want training to facilitate discussions with their patients about how they could talk to their children about a serious illness [17]. This is similar to the literature surrounding HCPs' beliefs and behaviours about talking with patients regarding spiritual and religious beliefs in clinical settings [18], despite the evidence that these conversations would be welcomed and beneficial to patients [19].

Research about communicating with children regarding a parent's physical illness has almost exclusively focused on cancer, HIV and palliative care [11]. Research has indicated that HCPs in palliative care often assume that the needs of children have been addressed at an earlier stage of the illness trajectory, yet parents report this has not happened [8, 20]. The early involvement of children when a parent has been diagnosed with a serious illness has been shown to be associated with better child outcomes when a parent reaches end of life care or dies [21]. Children themselves describe wanting more information about their parent's illness [22] but are usually reliant on their parents as the main source of information [23]. The healthcare team may be pivotal in encouraging communication within the family [14, 20, 23, 24]. Although there are published reviews and frameworks to support healthcare professionals with this task [1, 25] there are no service specifications in the United Kingdom (UK) regarding patient's children other than those related to safeguarding.

This study aimed to address a gap in the literature by recruiting HCPs from a range of different clinical specialties and disciplines working with patients with serious illness at any stage of the illness trajectory.

The current study explored HCPs' perceptions of their role in supporting patients to communicate with children about the diagnosis. The objectives were to explore HCPs':

- perceived role in having family-centred conversations in clinical care,

- understanding of parents' needs when they are diagnosed with a serious physical illness, and

- views on how to support families talk with children when a parent has a serious physical health condition.

## Materials and methods

### Design

A qualitative design using semi-structured interviews. The study is reported following the SRQR guidelines [26].

### Participants

Purposive and convenience sampling was used to recruit twenty-five HCPs from two NHS healthcare Trusts and two General Practices in the UK, of whom 24 were interviewed. Participants were eligible if they had clinical experience within the NHS caring for adult patients with serious illness such as cancer, lung disease or neurodegenerative conditions.

### Study recruitment

Participants were identified through established professional networks and snowball recruitment. Two authors (ER, LJD) sent an email (including the participant information sheet) inviting potential participants to take part in a one-off interview. Interested individuals replied to ER or LJD via email confirming they met the eligibility criteria, and to arrange a date and time for the interview. Written informed consent was obtained prior to the interview. There were nine non-responders, and one individual who later declined to be interviewed due to time commitments.

### Data collection

Semi-structured interviews were conducted between January and August 2019. A semi-structured interview guide was developed by ER and LJD, informed by the aims and objectives of the study as well as the prior experience of the research team, and pilot testing with clinicians in the study team. The guide was iteratively modified to enable follow-up with identified categories throughout the interview period (Table 1). Interviews were completed when no further categories were identified. Interviews took place in a private space, either face-to-face at the Department of Psychiatry, University of Oxford, or within the participant's work-place. Interviews were audio-recorded and lasted between 41 and 97 minutes (mean = 62 minutes). Interviews were completed by ER and LJD, neither of whom had prior relationships with the participants they interviewed; if a participant was known to one of the interviewers, the other researcher conducted the interview. To demonstrate reflexivity authors ER and LJD recorded reflections after each interview. These were discussed at regular team meetings throughout the data collection period.

### Data analysis

Audio-recordings were transcribed verbatim by a University approved transcription service and accuracy-checked by the research team. Reflexive thematic analysis guided the analysis [27, 28]. Initially, all authors read and re-read the transcripts to gain a sense of each participant's story. The second author [AM] coded the data using NVivo V11. Deployed as an inductive method, codes were developed by marking similar phrases or words in the narratives. AM undertook an analytical mind mapping process to explore patterns and relationships in the

**Table 1. Semi-structured interview guide.**

| Topics informed by the literature and the study aims and objectives | |
|---|---|
| Exploration of experiences of talking to adult patients who are parents about their serious illness | **Prompts**:<br>Examples of conversations that went well and why<br>Examples of conversations which were challenging and why |
| Exploration of HCPs' perceptions of parent's needs when a parent has a serious illness | |
| What support do parents with a serious illness need from professionals about their children | |
| Professionals' personal reflections of having conversations with parents about talking to their children about their serious illness | |
| **Additional topics as interviews progressed** | |
| Professionals' perceptions of who in the clinical team is best placed to have these conversations | |
| The emotional impact of supporting parents with a serious illness about their children | |

data and to identify themes [29]. These outputs were separately analysed by ER and LJD and, where there were differences, the third author was consulted [JRH]. Final themes were verified and refined through critical dialogue with all authors. To promote study trustworthiness, the findings were shared with three of the participants who felt the findings reflected their experiences.

## Ethical considerations

Professionals were provided with oral and written information about the study, and provided written consent. Participants were aware of the right to withdraw at any stage, data protection procedures were observed, and assurances of confidentiality were provided. Ethical approvals were obtained from Central University of Oxford Ethics Committee (R61317).

**Patient and public involvement statement.** No patients or members of the public were involved in the design of the study, nor in the interpretation of the results.

## Results and discussion

Overall, twenty-four HCPs were interviewed from a range of specialities, including primary care, critical care, respiratory, neurology, immunology, genetics, and oncology. Sample characteristics are reported in Table 2. Most professionals exclusively provided care to adult patients (n = 16), with a smaller number seeing mostly adult patients and some paediatric patients (n = 6) or working in General Practice who manage both adults and children (n = 2).

Results are discussed under three key themes identified in our findings with regards to family conversations about illness::

1. Identifying patients as parents; *knowing* as a prerequisite for acknowledgement and engagement

2. Healthcare professionals' perception of their roles and remits in relation to family conversations about illness

3. What do healthcare professionals need to initiate family-centred conversations?

**Table 2. Characteristics of the 24 participants interviewed in the study.**

| Healthcare Professional Speciality (Position) | Number |
| --- | --- |
| Clinical Genetics (Consultant) | 1 |
| Gastroenterology and Hepatology (Consultant) | 1 |
| General Medicine (Specialist Registrar) | 1 |
| General Practitioner (partner in their practice) | 2 |
| Immunology (Consultant) | 1 |
| Intensive Care (Consultant) | 2 |
| Neurology (Advanced Nurse Practitioner) | 1 |
| Neurology (Consultant) | 1 |
| Obstetrics and Gynaecology (Foundation Year 2 doctor) | 1 |
| Oncology (Advanced Nurse Practitioner) | 1 |
| Oncology (Clinical Nurse Specialist) | 3 |
| Oncology (Consultant) | 1 |
| Oncology Radiographer (band 7)* | 1 |
| Specialist Orthoptist (band 6)* | 1 |
| Radiotherapy (Clinical Nurse Specialist) | 1 |
| Respiratory and General Medicine (Consultant) | 1 |
| Respiratory (Specialist Registrar) | 1 |
| Rheumatology (Consultant) | 1 |
| Surgery (Consultant) | 1 |
| Trauma Physiotherapist (band 6)* | 1 |
| **Gender** | |
| Male | 9 |
| Female | 15 |

*NHS Agenda for Change pay scale band; reflects role and seniority of post holder

## Theme 01: Identifying patients as parents: *Knowing* as a prerequisite for acknowledgement and engagement

*Knowing* whether adult patients are parents is a pre-requisite for family-centred conversations; without this, the patient as a parent cannot be acknowledged and the topic of conversations with children cannot be engaged with. Some HCPs reported that they often did not know whether their patients had children and did not routinely ask patients (or their relatives) questions that might identify them as parents.

"*And do we ever do that [identify children whose parents are ill]? I don't think it happens, if I am honest.*"

[Participant 019, GP]

"It [patients' children] is not something you consciously think about."

[Participant 021, ITU]

Other HCPs suggested that parents would be identified through their clinical team's use of a proforma on admission to capture information about a patients' immediate family and social circumstances. A number of HCPs found that a patient's family relationships were indirectly identified through a 'patient concerns' screening tool, or generating a genogram as part of the clinical assessment where familial inheritance was a possibility. Despite HCPs reporting ways

of identifying and recording patient's children on their medical record, this did not necessarily translate to the implications of the illness for the wider family being acknowledged or addressed and it appeared most professionals did not follow up with patients about whether they had talked to their children about the diagnosis.

The pressure of time during clinical appointments was the main reason given for not asking about children's knowledge and understanding of the illness. Priorities were focused on disease and symptom management rather than conversations about any children, which were sometimes described as '*a family matter'* rather than a topic for medical consultations in adult services.

*"I mean you can't spend all day talking about stuff other than, you've got a lot to get through that's medical, that no one else can do."*

[Participant 005, Rheumatology]

At times, patients' parenting responsibilities were highlighted through a child's visit during an inpatient admission or their attendance at an adult's outpatient appointment. Differing attitudes were expressed by participants about responding to children's presence; some HCPs reported that if children had been brought to the hospital, they assumed parents were happy for the children to be aware of the clinical information being discussed. Conversely, others reported that they would suggest to parents that children should not be present during the consultation and to leave the room. Some participants expressed uncertainty about ascertaining young children's understanding of the situation which made them hesitant about children under the age of 10 years old being included in conversations:

*". . ..it is a serious conversation that you want to break to the family members who are, you know, of an age where they can understand appropriately……..I specifically ask for, you know, grandma and grandpa or friends, to just go over to Costa or something."*

[Participant 002, General Medicine]

## Theme 02: Healthcare professionals' perception of their roles and remits in relation to family conversations about illness

Participants described a range of opinions about who should take on the task of encouraging communication with children about a parent's serious illness. Within this, there were a range of approaches in terms of whether they perceived this to be appropriately within the boundaries of their role, or delegated or assumed it to be within another's remit.

Many HCPs did not perceive it was their role to initiate conversations with patients about what their children knew and understood about the illness. Some felt it was too intrusive to talk with patients about the children and this was a matter for parents alone, or that other organisations and charities were ideally placed to help parents.

*"Because you are talking to them [adult patient] directly, you tend to leave it to them to approach that [talking to their children]. Not that that is necessarily right, but it is the way it tends to happen."*

[Participant 015, Surgery]

*"I have asked 'who is at home?' but not really explored whether communication with children might be an issue or then what the patient is thinking about."*

[Participant 018, Oncology]

     

Professionals perceived that their patients were '*overwhelmed*' by the news of their diagnosis and believed it would be inappropriate to engage in conversations about sharing this news with the children at this stage.

> *"I wouldn't want to open up a conversation that perhaps the parent felt wasn't appropriate coming from me. I would never sit someone down and try to tell them how to deal with those situations. I don't want to tell them how to parent their children."*

[Participant 006, Oncology]

Some HCPs did not feel they were the right person to talk to the family regarding the children; these professionals expressed concern that they might cause distress to the parents by *"saying the wrong thing" [008, 018, 025]* or *"opening a can of worms" [012, 019]* without the skills to then address any problems identified.

> *"What can I say, what if they are in pieces, what do I do with that at that time of day? It seems a bit crass to enquire about something you aren't in a position to help with."*

[Participant 019, GP]

Medical staff often reported that having conversations with patients about the importance of sharing the diagnosis with their children was a clinical responsibility of general ward, community or specialist nurses within their team. This was attributed to the perceived closeness of nurses' relationships with patients and families, and nurses having "*more time'* to explore this with patients.

> *"They [clinical nurse specialists] have longer clinic slots in their clinics. And I think their role is the sort of liaison that means they do get closer to practicalities."*

[Participant 022, Respiratory]

However, reflecting on their perception of hierarchies within medicine and the high esteem in which doctors are held by patients, some HCPs (both consultants and nurses) felt that the medical consultant could play a key role in highlighting to parents the importance of talking to their children as part of the diagnostic consultation.

> *"In a nutshell, a comment from a consultant. If a consultant could say 'OK, so you have children, have you thought about how you're going to tell them?' It packs a punch. There is a lot of power in what is said in the conversation with the consultant. . . I think what they say is then taken on board."*

[Participant 006, Oncology]

Some HCPs working in an acute hospital setting described their assumption that another, often unspecified colleague, would be responsible for having conversations with parents about their children. However, none of these HCPs reported following-up with colleagues (or patients) to check if this support had been provided.

> *"How that is dealt with, I am not really in the loop. [. . .] So you don't really know what happens to those children in terms of who says what to them. Someone might, but not me."*

[Participant 015, Surgery]

*"I'm assuming, obviously, that is something they [family members] do, but I don't know if it's something specialist nurses do, and I would suspect that they would probably be more involved with that. I think there's even discussions or possibly a brochure. . . I don't know."*

[Participant 004, Gynaecology]

Many HCPs reported that patients did not talk to the healthcare team about their children; these HCPs assumed parents were receiving this support from other sources, such as their child's school or through charities. However, these professionals had not asked patients about how their children were coping or whether any additional help was needed regarding their children's understanding of their parent's illness.

*"I don't think I'm commonly asked. . ..there's a huge provision for welfare at school level and I suspect and imagine a lot of it is done at that level."*

[Participant 001, GP]

By contrast, there were a small group of HCPs that felt conversations about children should be an important element of routine practice, providing the rationale that clinical care should consider the needs of all the members of a patient's family. These HCPs felt that it was not in children's best interests to be *"kept in the dark"* about what was happening to their parent. The HCPs believed that information was beneficial for children in terms of preparing them for the next stage of the disease and ensuring that children did not feel a misplaced sense of responsibility for their parent's symptoms or physical changes.

*"I think it would be a dereliction of clinical care if you didn't try to think about the children and the family as a whole and the effect it is having on the family, in my view. People think it is not their job, but I mean of course it is."*

[Participant 010, Neurology]

These HCPs who clearly identified their own role in talking about children included both medical consultants and specialist nurses, working in oncology and neurology. All were routinely caring for patients who had definitive diagnoses associated with an extremely poor prognosis. The clinical nurse specialists from this group also reported that they would be willing to talk to children directly, but no parents had ever asked.

For those HCPs who felt that family-centred conversations were, or sometimes could be, part of their role, there were different perspectives on the types of information and support that patients might want or need. Within this were considerations for HCPS about positioning themselves as approachable without appearing to overstep the line and be deemed intrusive.

Some HCPs thought that parents did not want or need guidance about how to talk to the children about the parent's illness as they were navigating the situation themselves within the family. Others described offering some general support about family centred conversations, but felt if parents wanted specific guidance on talking to children then they would ask for it.

*"I tell them 'If you need any help let us know', but I don't give them anything specific and so I leave it to them. . .. . .I guess I don't engage in it unless they want it. I say 'we are here' and then let them come to us."*

[Participant 030, Gastroenterology and Hepatology]

More often, HCPs did not feel that their role was to talk with children directly; rather parents were considered 'better placed' as "*they know their children best*" [011]. There was a recognition that some parents might be reluctant to tell their children about the illness in order to protect them from distress.

Some nursing and medical staff that felt it was good practice to inform parents that it could be harmful not to tell their children about the illness as "*they will know something is wrong*". On occasion, HCPs encouraged parents to avoid vague language when sharing the diagnosis with the children such as '*a growth*', and to use *'simple'* language but specific terms associated with the illness such as '*cancer*'.

> "*I always say 'often children worry more if they feel that they don't understand or if something has been held back. . ...be open and honest but in an age-appropriate way'. . .*"
>
> [Participant 006, Oncology]
>
> "*Don't ever tell your child an untruth, don't say 'It's all going to be fine, I am going to live forever'. Be truthful.*"
>
> [Participant 011, Neurology]

Often, HCPs did not describe following up with patients at subsequent appointments about what children had been told, so they were unsure if anything (or what) had been communicated to the children about their parent's illness. There were some clinical nurse specialists that reported *'checking in'* with parents at later appointments about how conversations with their children about the illness had been. HCPs were aware that some families did not share the diagnosis with children, which they assumed was because parents found it too painful to disclose this news. There were also situations when HCPs felt concerned that children were privy to too much information about a parent's illness. For example, when they felt a younger adolescent was inappropriately present for an inpatient or outpatient discussion about treatment decisions. HCPs were uncertain about how to raise their concerns with the family about a child's involvement. Consequently, in these situations, professionals did not question the parent's decision to include the child in the consultations or what had (or had not) been shared with their children about the illness.

> "*Well actually I don't think you can ever say they need to know. Because she's* [the mum] *the one who has got to deal with the consequences when she gets home, it is up to her.*"
>
> [Participant 013, Critical Care]
>
> "*You can't not respect their decision, even if you don't think it is a good idea. You can't insist*"
>
> [Participant 021, ITU]

### Theme 03: What do HCPs need to initiate family-centred conversations?

Some HCPs described their own or colleagues' perceived concerns about having the skills and confidence to talk with parents about their children. These included their perceptions of junior colleagues' fears of "*using the wrong word, scaring somebody*" [017] and feeling unable to manage parental distress. Other professionals reported fearing that they would become upset themselves, with participants reflecting it to be '*emotionally easier*' to provide practical clinical tasks such as symptom management.

Many HCPs reported that they would like training about how to start conversations with parents about their children, with guidance about what language would be appropriate to use as well as how and when this can be incorporated into routine clinical appointments.

> "....give us some training, some insight into what not to say, what you absolutely should say, and if there are any do's and don'ts."

[Participant 007, Specialist Orthoptist]

HCPs' reflected on their wider experiences of observing colleagues in shaping their current practice around 'breaking bad news', although this rarely included initiating conversations with patients about their children. HCPs suggested that education sessions with colleagues who had more knowledge and experience around communicating with and about children would be beneficial in developing their skills to initiate family-centred conversations. A few HCPs described the benefits of observing senior colleagues initiating family-centred conversations, including junior staff being "*indoctrinated*" by the Lead consultants to include this as an aspect of their routine care.

> "I think it would be useful to have some guidance from professionals that have done it a lot...It is not something I have had to think about that much"

[Participant 022, Respiratory]

> "I have had no formal training on it and I would hate, you know, your worst fear is saying the wrong thing"

[Participant 025, Oncology]

A small number of HCPs described the benefits of family communication for patients' mental health and pain management. They expressed a desire for their colleagues to understand the advantages of this type of focussed conversation with parents about children.

> "It is really important for staff to know that actually there is so much evidence to show that if you have those types of conversations [about children] with patients....they feel fewer symptoms, because actually their real worries are being addressed...they've made a bit of a plan of how they are going to tell them..."

[Participant 012, Radiotherapy]

HCPs reflected on the emotional demands of talking to patients and their (adult) relatives about a diagnosis or prognosis. While breaking bad news was described as a routine part of HCPs' work, participants perceived it could be an additional challenge for their colleagues to think about raising the subject of a patient's children.

> "[HCPs] can't take on any more emotionally, and having that conversation, asking 'have you thought about how you might tell the children?'; it is very painful to have those conversations and I think people have a sense of 'I sort of want to preserve myself'."

[Participant 012, Radiotherapy]

Considering the needs of a patient's children were perceived to be personally more difficult when HCPs identified similarities between themselves and the patient, such as *'what if this happened to me or my family'*.

*"This lady was close to my age and we had children of similar ages, so all the time in your head you are thinking 'This could be me'. And like could I imagine having that conversation with my son, I think it does just make you a bit more, I don't know, I have felt more emotionally attached maybe."*

[Participant 025, Oncology]

HCPs felt that emotional support from colleagues was needed to manage these difficult situations and highlighted the importance of opportunities to '*offload*' after sharing difficult diagnoses or prognoses with patients. A few participants mentioned the need for support from their partners at home to cope with the emotional aspects of their role.

## Discussion

Talking to patients about their diagnosis or prognosis was part of all the participants' professional roles; nonetheless, many reflected on the inherent emotional demands of this task. HCPs were uncertain about their role in having conversations with patients about their children, or even what support they could offer parents to talk to children about parental illness. During diagnostic consultations, most HCPs described the existence of reporting systems which could identify a patient's family network. However, information about the family was rarely used to initiate discussions with parents about their children's knowledge or understanding of the illness. This was often assumed to be addressed by someone else in the healthcare system, most frequently nursing staff. Participants reported lacking the specific skills and training to facilitate family centred conversations about a patient's diagnosis as well as time constraints and perceived remits of the clinical consultation.

Many HCPs reflected that communication about illness with children was a family matter and expressed uncertainty about whether patients would welcome them raising this topic. Crucially, research indicates that these concerns are unfounded; parents consistently report wanting support and guidance from HCPs about how to talk to their children [10, 20, 30, 31]. Parents need: (1) reassurance that telling their children is beneficial, and (2) the opportunity to discuss how to initiate conversations with their children [8, 9, 11, 24]. In the context of the ever-increasing pressures on the medical system, this raises the question of **who** is best placed to support parents with these tasks, and **how** this can be prioritised amidst the multiple demands on time-limited appointments.

Many participants assumed that colleagues elsewhere in their patients' care pathway were helping families tell children about the diagnosis. This suggests that there may be a risk of everyone within the healthcare system supposing that somebody else is identifying patients' children and their needs. This highlights the need for a systematic approach to family-centred conversations to ensure that all patients are given the opportunity for the support they may need to talk to their children. Clear documentation of these discussions in patients' notes may prompt the clinical care team to review children's understanding as the parents' illness changes. Empowering parents to talk to their children about the diagnosis may be promoted by the unique position of medical consultants in the patient-doctor relationship.

The role of HCPs in identifying children who may be affected by parental illness has been recognised and enshrined in law in Sweden and Norway [32]. This law requires HCPs to register dependent children in the patient's health record, as well as providing advice and guidance to parents about supporting their children through the illness experience [13, 33, 34].

Medical consultations are rightly focused on the needs and priorities of the adult patient. Children are rarely brought to outpatient appointments, exacerbating the invisibility of a patient's wider family network and particularly relationships with children. This became

acutely evident during the COVID-19 pandemic due to visiting restrictions [20, 35]. Furthermore, patients report feeling uncertain about whether it is appropriate to raise concerns with the healthcare team about how to tell their children [30, 36, 37]. Parents' hesitation may inadvertently reinforce HCPs' perception that patients consider talking to children as a '*family matter*' and that they do not want any support around this unenviable task [20]. Wider dissemination of the evidence about the benefits for adult patients of effective communication with children may help to raise the priority of this agenda item for healthcare consultations. This understanding may also help reassure both patients and HCPs that they can employ practical steps to mitigate the psychological impact of parental illness on children.

A lack of confidence and skills were contributing factors towards HCPs' hesitation about initiating conversations with patients about their children. A structured, evidence-based framework may help HCPs plan and undertake these discussions with their patients [1, 31, 35]. These frameworks include guidance for HCPs about how to explore parents' views about talking to their children; common reactions, questions and concerns from children with suggested responses; and recommended language and phrases that professionals can use when having conversations with parents who have a life-threatening condition about their children. The development of specific video-based resources of HCPs talking to patients about telling children a parent's diagnosis may be a valuable and accessible tool to address HCPs training needs [24].

## Strengths and weaknesses of the study

This study explored a range of HCPs' experiences of communication, including participants from a range of medical specialties, professional backgrounds, and seniority, working in both acute and community settings, although not every specialty or professional background is included. The study complements existing work focusing on family-centred care at End of Life by enhancing our understanding about how and when patients are identified as parents at an earlier stage and across healthcare. The recruitment strategy of drawing on pre-existing professional networks and snowball recruitment may have introduced bias into the participant sample. The study is also limited by not including the experiences of multiple HCPs from different disciplines contributing to the same care pathway or from a wider range of geographical NHS trusts.

Future research should consider how to address the challenge of integrating family centred conversations into routine patient care and the specific needs of different professional groups and healthcare settings. There is a need to develop and evaluate accessible and feasible resources to enhance HCPs' confidence and skills in raising these sensitive conversations with their patients.

## Conclusion

When a parent has a serious illness, the needs of children must not be overlooked. Healthcare professionals treating adult patients should have a clear understanding of the pivotal role they can play in supporting parents to talk to their children about a diagnosis. Evidence indicates that such guidance would be welcomed by patients, with long-term benefits for children's psychological health and the burden on mental health services.

## Acknowledgments

The authors are indebted to all of the participants for sharing their experiences so generously. Our thanks to Professor Sue Ziebland and Dr Simone de Cassan for their thoughtful comments on the manuscript.

## Author Contributions

**Conceptualization:** Louise J. Dalton, Elizabeth Rapa.

**Data curation:** Louise J. Dalton, Elizabeth Rapa.

**Formal analysis:** Louise J. Dalton, Abigail McNiven, Jeffrey R. Hanna, Elizabeth Rapa.

**Funding acquisition:** Louise J. Dalton, Elizabeth Rapa.

**Investigation:** Louise J. Dalton, Elizabeth Rapa.

**Methodology:** Louise J. Dalton, Elizabeth Rapa.

**Project administration:** Louise J. Dalton, Elizabeth Rapa.

**Writing – original draft:** Louise J. Dalton, Abigail McNiven, Elizabeth Rapa.

**Writing – review & editing:** Louise J. Dalton, Abigail McNiven, Jeffrey R. Hanna, Elizabeth Rapa.

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
