## [Decision Letter · Decision Letter 0]

9 Aug 2022

PONE-D-22-10150Exploring healthcare professionals’ beliefs and experiences of family-centred conversations when a parent has a serious illness: a qualitative studyPLOS ONE

Dear Dr. Dalton,

Thank you for submitting your manuscript to PLOS ONE. After careful consideration, we feel that it has merit but does not fully meet PLOS ONE’s publication criteria as it currently stands. Therefore, we invite you to submit a revised version of the manuscript that addresses the points raised during the review process.

Your manuscript has been reviewed by one peer-reviewer. They commented that your manuscript can be strengthened by improvements to the reporting of this study, for example the recruitment procedure, the interview guide, and the analysis and reporting of the results. The full review is found at the bottom of this email.     Could you please revise the manuscript to carefully address the concerns raised? Please note that we have only been able to secure a single reviewer to assess your manuscript. We are issuing a decision on your manuscript at this point to prevent further delays in the evaluation of your manuscript. Please be aware that the editor who handles your revised manuscript might find it necessary to invite additional reviewers to assess this work once the revised manuscript is submitted. However, we will aim to proceed on the basis of this single review if possible. 

We look forward to receiving your revised manuscript.

Kind regards,

Maria Elisabeth Johanna Zalm, Ph.D

Editorial Office

PLOS ONE

Journal Requirements:

Reviewers' comments:

Reviewer's Responses to Questions

**Comments to the Author**

1. Is the manuscript technically sound, and do the data support the conclusions?

Reviewer #1: Partly

2. Has the statistical analysis been performed appropriately and rigorously? 

Reviewer #1: N/A

3. Have the authors made all data underlying the findings in their manuscript fully available?

Reviewer #1: Yes

4. Is the manuscript presented in an intelligible fashion and written in standard English?

Reviewer #1: Yes

5. Review Comments to the Author

Reviewer #1: PONE-D-22-10150 Exploring healthcare professionals’ beliefs and experiences of family-centered conversations when a parent has a serious illness: a qualitative study

1. The study presents the results of original research.

Yes, this is original research.

2. Results reported have not been published elsewhere.

Yes

3. Experiments, statistics, and other analyses are performed to a high technical standard and are described in sufficient detail.

High technical standard and in sufficient detail, but there is a need of clarification about the recruitment procedure, which is purposive, convenience, snowball sampling via established professional network? Emails were sent out, to whom? Please, give us some more information.

The aim was to address a gap in the literature by recruiting HCPs from a range of different clinical specialties and disciplines working with patients with serious illness at any stage of the illness trajectory.

The aim and research question are formulated for a content analysis, re-formulate this to be in line with the theoretical framework presented- thematic analysis.

The title is about beliefs and experiences of family-centred conversations and the participants are working mostly with adults and in the introduction, you state that HCPs report a lack of adequate knowledge about children’s developmental understanding or age-appropriate communication to have conversations with patients about their children.

So, is this study mostly about beliefs and opinions?

Face-to-face interviews took place at the Department of Psychiatry, University of Oxford or within the participant’s workplace. There was no disturbance?

Could you please, present the interview guide. How many questions/topics were there? This was modified during the interview trajectory. Interviews were completed when no further categories were identified. Did you make parallel data collection and analysis? There seem to be influences of Grounded theory.

Since this is a qualitative study “exploring” HCP experiences it would have been interesting and useful to know age and years of professional experience of the HCPs

Trustworthiness is presented, by participants (3) confirming the result, quotations, but only for the analysis and the findings. No thoughts about the audit trail.

What are these quotations presented below the participant table?? Some kind of result but not a theme??

There were 24 interviews conducted, but the quotations are numbered 1-25, how come?

In the result there are quotations only from 14 participants, but several quotations from about 4-5 participants.

The result is five 3 themes. There are 3 participants quotations in the first theme, 12 participants quotations in the second one and three participants quotations in the last theme. Braun and Clarke state that; Our conceptualisation of themes – as stories about particular patterns of shared meaning across the dataset – is confused with ‘domain summaries’ – summaries of the range of meaning in the data related to a particular topic or ‘domain’ of discussion. The themes presented here are description, summarized text. Interpretation is inherent to the (TA) analytic process, and there is nothing in the method of TA that renders it simply summative or descriptive. There is a need for re-analyse the data, there should probably be two well balanced themes.

This result does not fulfil the aim or answering the research questions.

Sorry, but here is no analysis and interpretation- there is labels and then quotations piled, but not all participants are represented.

This study is focusing on an important and interesting area of research, so if the analysis were really reflexive there should have been more information-stories told, and there could have been a very good discussion if there was an interested and evidence-based result to discuss.

Limitations are lacking methodology issues. Was this study really focusing on a diverse range of HCPs’ experiences of communication?

4. Conclusions are presented in an appropriate fashion and are supported by the data.

Conclusions are presented in appropriate fashion, probably supported by data if re-worked.

There is no interpretation, so looking at the results there is a content analysis performed. Data is described and explored.

5. The article is presented in an intelligible fashion and is written in standard English.

Yes, the article is presented in an intelligible fashion, and it is written in standard English.

6. The research meets all applicable standards for the ethics of experimentation and research integrity.

Yes, this study meets all the applicable standards for research integrity.

7. The article adheres to appropriate reporting guidelines and community standards for data availability.

Yes, the article is following the reporting guidelines.

This could be an interesting paper, presenting important knowledge. This paper needs to be re-worked, re-analysed and re-written presented with correct methodology then it would be suitable for publication.

Out of 37 references 11 are 10 years old or more, there are 25 references in the introduction and out of these 7 are 10 years old or more.

6. PLOS authors have the option to publish the peer review history of their article (what does this mean?). If published, this will include your full peer review and any attached files.

Reviewer #1: No

---

## [Author Response · Author response to Decision Letter 0]

6 Oct 2022

Exploring healthcare professionals’ beliefs, experiences and opinions of family-centred conversations when a parent has a serious illness: a qualitative study

Response to reviewer’s comments

We thank the reviewer for their comments about the manuscript, and their recognition that this is “an interesting paper, presenting important knowledge”. We have made revisions to the manuscript (shown as tracked changes) as requested. Point by point responses to the reviewer’s comments are given below.

Reviewer 1: High technical standard and in sufficient detail, but there is a need of clarification about the recruitment procedure, which is purposive, convenience, snowball sampling via established professional network? Emails were sent out, to whom? Please, give us some more information.

We have re-written the Materials and Methods section, including the addition of a Study Recruitment subsection, to expand and make these details clearer to the reader.

Reviewer 1: The aim was to address a gap in the literature by recruiting HCPs from a range of different clinical specialties and disciplines working with patients with serious illness at any stage of the illness trajectory. The aim and research question are formulated for a content analysis, re-formulate this to be in line with the theoretical framework presented- thematic analysis. 

The aims and objectives are set in the context of the background literature and the gaps in the current literature i.e. this study aimed to address a gap in the literature by recruiting HCPs from a range of different clinical specialties and disciplines working with patients with serious illness at any stage of the illness trajectory. 

The current study explored HCPs’ perceptions of their role in supporting patients to communicate with children about the diagnosis. The objectives were to explore HCPs’: 

• perceived role in having family-centred conversations in clinical care, 

• understanding of parents’ needs when they are diagnosed with a serious physical illness, and 

• views on how to support families talk with children when a parent has a serious physical health condition. 

Thematic analysis was used as part of the analysis strategy to understand the broader themes within the data; in sum, our approach was to collect the experiences and perspectives of healthcare professionals, in order to identify and interpret the meanings and values they express, and build a comprehensive understanding of their experiences. Content analysis was not part of this study. 

Reviewer 1: The title is about beliefs and experiences of family-centred conversations and the participants are working mostly with adults and in the introduction, you state that HCPs report a lack of adequate knowledge about children’s developmental understanding or age-appropriate communication to have conversations with patients about their children. So, is this study mostly about beliefs and opinions?

The study is exploring participants’ beliefs and opinions, as embedded in their experiences of engaging (to varying degrees) with family-centred conversations in their professional lives. As such, HCPs beliefs and opinions are part of this, and, as such, we have added ‘opinions’ to the title. 

Reviewer 1: Face-to-face interviews took place at the Department of Psychiatry, University of Oxford or within the participant’s workplace. There was no disturbance?

We have added ‘private space’ to the data collection section to make it clear that the interviews took place with no disturbances.

Reviewer 1: Could you please, present the interview guide. How many questions/topics were there? This was modified during the interview trajectory. Interviews were completed when no further categories were identified. Did you make parallel data collection and analysis? There seem to be influences of Grounded theory. 

The interview guide has now been added as Table 1. The topic guide was updated during data collection as additional topics were identified during data collection that warranted follow-up during subsequent interviews. The topic guide covered areas such as: Exploration of experiences of talking to adult patients who are parents about their serious illness, Exploration of HCPs’ perceptions of parent's needs when a parent has a serious illness, What support do parents with a serious illness need from professionals about their children, Professionals' personal reflections of having conversations with parents about talking to their children about their serious illness. Additional topics were: Professionals’ perceptions of who in the clinical team is best placed to have these conversations, The emotional impact of supporting parents with a serious illness about their children.

Reviewer 1: Since this is a qualitative study “exploring” HCP experiences it would have been interesting and useful to know age and years of professional experience of the HCPs

In Table 2, we have stated the position of each HCP which reflects the experience of the HCP. We do not wish to provide any further information about each HCP, as we feel this would jeopardize confidentiality. We report participants’ demographic and personal circumstance details (such as having children of their own) only where the participant themselves deemed it relevant to their experiences or perspectives.

Reviewer 1: Trustworthiness is presented, by participants (3) confirming the result, quotations, but only for the analysis and the findings. No thoughts about the audit trail.

Our findings were shared with three participants from the study, who felt that these reflected and summarised their experiences accurately. 

Regarding the audit trail, we have added further clarification to the Data Collection section of the Methods.

Reviewer 1: What are these quotations presented below the participant table?? Some kind of result but not a theme??

In response to the reviewer’s comment about presenting the themes differently we have now incorporated this whole section into theme 3.

Reviewer 1: There were 24 interviews conducted, but the quotations are numbered 1-25, how come?

We have re-worded this section in the methods, stating that 25 professionals were recruited into the study and allocated participant numbers but one individual later declined to take part in an interview due to time commitments (as stated in the methods). As such, quotations will be from participants numbered 1-25.

Reviewer 1: In the result there are quotations only from 14 participants, but several quotations from about 4-5 participants. The result is five 3 themes. There are 3 participants quotations in the first theme, 12 participants quotations in the second one and three participants quotations in the last theme. Braun and Clarke state that; Our conceptualisation of themes – as stories about particular patterns of shared meaning across the dataset – is confused with ‘domain summaries’ – summaries of the range of meaning in the data related to a particular topic or ‘domain’ of discussion. The themes presented here are description, summarized text. Interpretation is inherent to the (TA) analytic process, and there is nothing in the method of TA that renders it simply summative or descriptive. There is a need for re-analyse the data, there should probably be two well balanced themes.

The goal of this study was not to develop a thematic framework for analysis, but rather the transcripts were individually coded and through the inductive method where codes were developed by marking similar phrases or words in the narratives, followed by analytical mind mapping to explore patterns and relationships in the data to identify themes. Final themes were verified and refined through critical dialogue with all authors. This is in line with reflective thematic analysis (Braun and Clarke 2019) as stated in the data analysis of the manuscript. 

Our intended audience for the paper includes healthcare professionals and, with recognition of their need to have clearly signposted findings, we structured our paper and presented our findings with relatively descriptive headings. The findings presented are based on our thematic analysis, but our presentation in the paper is intended to have a pragmatic focus in terms of relevance for healthcare professionals and other time-constrained stakeholders with an interest in family-centred clinical conversations. We do not agree that re-analysis is necessary, however we have taken on board the need to communicate the phrasing of our findings in a different way and have substantially reworked sections of the results sections. 

In response to the reviewer’s feedback concerning participant’s quotations, we have changed a quote from 025 to a quote from 021, and added quotes from 011, 030 and 007, in order to include the voices of different participants. In summary, quotes from 19 out of the 24 participants have now been included in the manuscript. The quotes selected are illustrative of experiences, including those which were shared across a number of participants or which were relatively unique amongst our participants, and therefore quote selection is not intended to be an exhaustive representation of all participants in the study.

Reviewer 1: This result does not fulfil the aim or answering the research questions. Sorry, but here is no analysis and interpretation- there is labels and then quotations piled, but not all participants are represented. This study is focusing on an important and interesting area of research, so if the analysis were really reflexive there should have been more information-stories told, and there could have been a very good discussion if there was an interested and evidence-based result to discuss. 

As highlighted previously, the way we have structured the paper and phrased subsections is intended to aid engagement from healthcare professionals. We recognise that this has particular implications for the communication of qualitative research, and have sought to strike a balance between accessibility and the richness of experience as expressed in participant quotes. This has included expanding and restructuring parts of the results sections.

Within our findings, we have thus positioned our narrative in relation to our study aims. We have extensively reported HCPs’ different perceptions of their role in having family-centred conversations in clinical care in theme 1 (some HCPs feel it is not a priority for the consultation, and purposefully decide to exclude children from a consultation; ) and theme 2 (someone else in the team will have this conversation). We report HCPs’ understanding of what parents need in theme 2 (such as their belief that the family are navigating the process themselves or accessing support from other sources e.g. school). Views on how to support families talking to children is reported in theme 3, highlighting HCPs desire for further training and skill development opportunities. In each case, the patterns of similarities and differences in participant experiences and perspectives are explored.

This is reflective of the individual interviews, which is reflected within the topic guide, designed to achieve the aims and objectives of the study. 

Reviewer 1: Limitations are lacking methodology issues. Was this study really focusing on a diverse range of HCPs’ experiences of communication?

We have deleted ‘diverse’ from the sentence, with recognition that the breadth of HCPs is extremely wide, but reiterate that a range of participants from different professionals and specialties were sought. We have added a clarification that ‘not every specialty and professional background is included.’

Reviewer 1: Out of 37 references 11 are 10 years old or more, there are 25 references in the introduction and out of these 7 are 10 years old or more. 

In the introduction, we have added Ellis et al. 2017 and removed reference Helseth et al. 2003 and Visser et al. 2004. The remaining papers which are 10 years old or more represent the work of Fallowfield, Semple and Fearnley who are still world leading experts in this area and the papers referenced are seminal pieces of work in this area of research. In addition, Rauch et al. 2002 is one of the first papers to highlight the importance of raising the issue of patient’s children and thus must be acknowledged. Furthermore, the age of the literature cited signals that this topic has not been explored further in recent years, and reiterates why our study is necessary and timely.

Reviewer 1: Conclusions are presented in appropriate fashion, probably supported by data if re-worked.

There is no interpretation, so looking at the results there is a content analysis performed. Data is described and explored. 

In the conventions of reporting qualitative findings, we have used the results section to describe the data in line with the aims and objectives, and present the results in story-narrative as per reflexive thematic analysis (Braun and Clarke 2019). Our interpretation of the data, and the wider context of implications, is described in the discussion. For example, we link our results detailing HCPs’ reported opinions of communication with children with the wider literature from parents who want help and guidance from their clinical team in talking to their children. We discuss the clinical implications of the routine absence of identifying children by clinical teams (results of theme 1) and how this could be addressed in practice. The discussion also builds on the results of theme 3 by outlining ways to address the needs of HCPs to facilitate family-centred conversations.

Reviewer 1: This could be an interesting paper, presenting important knowledge. This paper needs to be re-worked, re-analysed and re-written presented with correct methodology then it would be suitable for publication.

We are grateful to the reviewer for highlighting areas for improvement; we have made significant changes to the methods and results sections based on their feedback. We look forward to hearing your thoughts on the re-worked paper.

---

## [Decision Letter · Decision Letter 1]

4 Nov 2022

PONE-D-22-10150R1Exploring healthcare professionals’ beliefs, experiences and opinions of family-centred conversations when a parent has a serious illness: a qualitative studyPLOS ONE

Thank you for submitting your manuscript to PLOS ONE. After careful consideration, we feel that it has merit but does not fully meet PLOS ONE’s publication criteria as it currently stands. Therefore, we invite you to submit a revised version of the manuscript that addresses the points raised during the review process.

We look forward to receiving your revised manuscript.

Kind regards,

Luigi Lavorgna

Academic Editor

PLOS ONE

Journal Requirements:

Reviewers' comments:

Reviewer's Responses to Questions

**Comments to the Author**

1. If the authors have adequately addressed your comments raised in a previous round of review and you feel that this manuscript is now acceptable for publication, you may indicate that here to bypass the “Comments to the Author” section, enter your conflict of interest statement in the “Confidential to Editor” section, and submit your "Accept" recommendation.

Reviewer #1: All comments have been addressed

Reviewer #2: All comments have been addressed

2. Is the manuscript technically sound, and do the data support the conclusions?

Reviewer #1: Yes

Reviewer #2: Yes

3. Has the statistical analysis been performed appropriately and rigorously? 

Reviewer #1: N/A

Reviewer #2: Yes

4. Have the authors made all data underlying the findings in their manuscript fully available?

Reviewer #1: No

Reviewer #2: Yes

5. Is the manuscript presented in an intelligible fashion and written in standard English?

Reviewer #1: Yes

Reviewer #2: Yes

6. Review Comments to the Author

Reviewer #1: Thank you for all efforts, most of the comments have been acknowledged. The manuscript has been amended and clarified.

Even though I still think that there should be 2 balanced themes instead of three. The thirted one is very weak and seems not to fulfill the idea ofd Braun and Clark; Themes are creative and interpretive stories about the data, produced at the intersection of the researcher’s theoretical assumptions, their analytic resources and skill, and the data themselves

Just some comments, Braun and Clarke states that there often are some missinterpretations of TA;... our approach is ‘supplemented’ with other analytic procedures (your ref 27), sometimes because our approach on its own is deemed not ‘sophisticated’ enough for anything other than (often atheoretical) data description or summary (of surface meaning).(themes 1 and 2). A second comment from Braun and Clarke regarding missuse is;... Grounded theory concepts and procedures are attributed to TA.

Audit trail is about being able to following all steps during the research procedures and reflexitivy is about being aware own fore sights and assumptions and how that will influence analysis and results.

Just as an information, for further studies

Reviewer #2: Thank you for all correction that you made to your manuscript. I have an other suggestion to better improve your paper.

You say in introduction "HCPs’ report concerns that starting conversations

with patients about their children will not be welcomed or create distress for the patient." Do you know other distress argument for the patient? For example also talking about religiosity is a distress for patient but some study shown that although it represent a distress is important to talk about it (Sparaco M, Miele G, Abbadessa G, Ippolito D, Trojsi F, Lavorgna L, Bonavita S. Correction to: Pain, quality of life, and religiosity in people with multiple sclerosis. Neurol Sci. 2021 Dec 10. doi: 10.1007/s10072-021-05814-x. Epub ahead of print. Erratum for: Neurol Sci. 2021 Nov 23;: PMID: 34890003. ). Please discuss it.

7. PLOS authors have the option to publish the peer review history of their article (what does this mean?). If published, this will include your full peer review and any attached files.

Reviewer #1: No

Reviewer #2: No

---

## [Author Response · Author response to Decision Letter 1]

8 Nov 2022

Response to reviewer’s comments

We are grateful to the reviewers for considering our revised manuscript and are delighted that they are happy with our revisions.

Reviewer 1:

Thank you very much for your helpful information regarding Braun and Clarke’s methodology and will be sure to take this into account in our future work.

Reviewer 2:

 Thank you for highlighting the importance of religious and spiritual beliefs and the parallels with talking to patients about their family network. We have added the suggested reference of Sparaco et al., 2022, as well as a systematic review regarding this important literature (Best et al., 2016)

---

## [Editor Report · Decision Letter 2]

10 Nov 2022

Exploring healthcare professionals’ beliefs, experiences and opinions of family-centred conversations when a parent has a serious illness: a qualitative study

PONE-D-22-10150R2

We’re pleased to inform you that your manuscript has been judged scientifically suitable for publication and will be formally accepted for publication once it meets all outstanding technical requirements.

Kind regards,

Luigi Lavorgna

Academic Editor

PLOS ONE
---

## [Editor Report · Acceptance letter]

15 Nov 2022

PONE-D-22-10150R2 

Exploring healthcare professionals’ beliefs, experiences and opinions of family-centred conversations when a parent has a serious illness: a qualitative study 

Dear Dr. Dalton:

I'm pleased to inform you that your manuscript has been deemed suitable for publication in PLOS ONE. Congratulations! Your manuscript is now with our production department. 

Kind regards, 

on behalf of

Dr. Luigi Lavorgna 

Academic Editor

PLOS ONE